# Morphological Characteristics of the Double Mental Foramen and Its Relevance in Clinical Practice: An Observational Study

**DOI:** 10.3390/diagnostics14121277

**Published:** 2024-06-17

**Authors:** Alejandro Bruna-Mejias, Pablo Nova-Baeza, Florencia Torres-Riquelme, Maria Fernanda Delgado-Retamal, Mathias Orellana-Donoso, Alejandra Suazo-Santibañez, Walter Sepulveda-Loyola, Iván Valdés-Orrego, Juan Sanchis-Gimeno, Juan José Valenzuela-Fuenzalida

**Affiliations:** 1Departamento de Ciencias y Geografia, Facultad de Ciencias Naturales y Exactas, Universidad de Playa Ancha, Valparaiso 2360072, Chile; alejandro.bruna@upla.cl; 2Unidad de Anatomía Humana Normal, Escuela de Medicina, Facultad de Ciencias Médicas, Universidad de Santiago de Chile, Santiago 9160000, Chile; 3Departamento de Ciencias Químicas y Biológicas, Facultad de Ciencias de la Salud, Universidad Bernardo O’Higgins, Santiago 8370993, Chile; 4Departamento de Morfología, Facultad de Medicina, Universidad Andrés Bello, Santiago 8370146, Chile; ftorresriquelme@gmail.com (F.T.-R.); fernanda.delgado1771@gmail.com (M.F.D.-R.); juan.kine.2015@gmail.com (J.J.V.-F.); 5Escuela de Medicina, Universidad Finis Terrae, Santiago 7501015, Chile; mathor94@gmail.com; 6Department of Morphological Sciences, Faculty of Medicine and Science, Universidad San Sebastián, Santiago 8420524, Chile; 7Faculty of Health and Social Sciences, Universidad de Las Américas, Santiago 8370040, Chile; alej.suazo@gmail.com (A.S.-S.); walterkine2014@gmail.com (W.S.-L.); 8Facultad de Ciencias de la Salud, Universidad Autónoma de Chile, Santiago 8910060, Chile; ivan.valdes@gmail.com; 9GIAVAL Research Group, Department of Anatomy and Human Embryology, Faculty of Medicine, University of Valencia, 46001 Valencia, Spain; juan.sanchis@uv.es

**Keywords:** mental foramen, anatomical variation mental foramen, double mental foramen, clinical anatomy

## Abstract

The mental foramen (MF) is an opening found bilaterally on the anterolateral aspect of the mandible; it can be round or oval and have different diameters. One of the anatomical variants of the jaw is the presence of an accessory mental foramen (AMF). These are usually smaller than the MF and can be located above, below, or to the sides of the main MF. The objective of this study was to recognize the presence of AMF in dry jaws of the Chilean population and collect information about its clinical relevance reported in the literature. In this descriptive observational study, we have collected dried jaws obtained from three higher education institutions in Santiago de Chile, from the Department of Morphology of the Andrés Bello University, the Normal Human Anatomy Unit of the University of Santiago, and the Human Anatomy pavilion from the Faculty of Medicine of the Finis Terrae University. The samples for this research were obtained by convenience, and the observation of the jaws was carried out in the human anatomy laboratories of each institution by three evaluators independently, and a fourth evaluator was included to validate that each evaluation was correct. The sample for this research came from 260 dry jaws, showing the following findings from the total jaws studied, and to classify as an accessory MF, it will be examined and measured so that it complies with what is declared in the literature as the presence of AMF, which is between 0.74 mm. and 0.89 mm. There were 17 studies included with a sample that fluctuated between 1 and 4000, with a cumulative total of 7946 and an average number of jaws analyzed from the studies of 467.4, showing statistically significant differences between the means with the sample analyzed in this study; *p* = 0.095. For the cumulative prevalence of the presence of AMF, this was 3.07 in this study, and in the compared studies, the average of AMF was 8.01%, which did not present a statistically significant difference; *p* = 0.158. Regarding the presence of variants of unilateral AMF, this occurred in five jaws, which is equivalent to 1.84% in the sample of this study, while in previous studies, it was 7.5%, being higher on the left side than on the right. The presence of AMF is a variant with high prevalence if we compare it with other variants of the jaw. Knowledge of the anatomy and position of the AMF is crucial to analyze different scenarios in the face of surgical procedures or conservative treatments of the lower anterior dental region.

## 1. Introduction

The mental foramen (MF) is an opening found bilaterally on the anterolateral aspect of the mandible; it can be round or oval and have different diameters [1]. One of the anatomical variants of the mandible is the presence of accessory mental foramen (AMF). These are usually smaller than the main MF and are located either above, below, or to the sides of the main MF [2]. The presence of AMF can be both bilateral and unilateral, singular, or multiple, which is uncommon. The characteristic of this foramen is that it is connected to the mandibular canal; given that any other orifice that is present on the anterior face of the mandible near the MF that is not in relation to the mandibular canal, it is considered a nutritive foramen, which usually tends to be smaller in size than the AMF and MF [3]. This is relevant because the content of this AMF is the same as the main MF, that is, the inferior alveolar artery, inferior alveolar nerve, mental nerve, and inferior alveolar vein. According to the aforementioned information, we can affirm that the content of the MF is identical to that of the AMF. Among other vessels and neurovascular structures, these structures innervate the gums, teeth, the inner surface of the cheeks, the skin of the chin, the skin of the corners of the mouth, the mucous membrane, and the lower lip, in addition to irrigating the entire area surrounding these structures [4].

From the point of view of embryology, the MF does not complete its development until the 12th week of gestation, which happens after the mental nerve branches into fascicles. The jawbone begins its formation once it is found. The inferior alveolar nerve and its branches are complete, so the MF and the MFA are formed in the same time range. Once the mandible has matured, no more MFAs are formed. An example of this is Balcioglu’s study from 2011, which describes a 30-week fetus with an MFA on the right side [5].

The range of prevalence in the population varies according to ethnicity and sex, ranging from 1% in the Russian population to 10% in the Arabic population. The different studies systematically reviewed confirm a lower presence of this foramen in Caucasian populations and a greater predominance in Middle Eastern countries [3,6,7,8]. This takes on special relevance when performing dental procedures since, although in some populations it is very rare for the patient to present this anomaly as in countries like Saudi Arabia, 1 in 10 patients have it. The sex of the patient also becomes relevant since the majority of studies reviewed agree that its presence is greater in men than in women, and its highest incidence is on the right side of the jaw [3]; however, women are more likely to have multiple AMFs than men, according to the Gümüşok study, in which only women had multiple AMFs [9]. This variant originates in a branch of the mental nerve before it exits through the MF, thus resulting in the different locations, sizes, and arrangements of this AMF [1,5].

In terms of clinical complications, it usually causes problems in anesthesia procedures, parasymphyseal fractures, or fractures that affect the jaw near the area of the first and second premolar, genioplasty treatment, orthognathic surgeries, dental implant surgeries, osteotomy, neurectomy, trauma, mandibular subapical surgery, periapical surgeries, open reduction of an anterior mandibular fracture, inferior alveolar nerve transposition surgery, and molar extraction [1,10]. In the case of local anesthesia, what is sought is the mental nerve block; however, if the distance between the AMF and the MF is wide, a double anesthesia must be performed to prevent the patient from feeling pain since the AMF presents an accessory branch of the mental nerve [11,12]. In the case of implants, the location of the AMF becomes especially important, because its arrangement limits the length and width of the dental implant to be placed in the interforaminal area, which is why it is important to calculate the distance between the AMF and MF [13]. In the case of genioplasty, this variously causes an intervention in the osteotomy lines that are drawn to perform the surgery, interfering in a similar way in mandibular subapical surgery. On the other hand, in transposition surgery of the inferior alveolar nerve, it generated greater difficulty in performing surgery [14]. If this is not taken into consideration, various iatrogenic injuries can be inflicted on the patient, ranging from numbness of the lower lip, hypoesthesia of the mandibular area, cheeks, chin, corners of the mouth, and lips, to hemorrhages or permanent damage to the accessory mental nerve [15]. Therefore, it is important to complete an evaluation before performing any procedure on the symphysis and mandibular body or in case the patient presents recurrent problems with anesthesia. This should be performed using a cone beam computed tomography (CBCT), which will provide a precise 3D image of the mandible, thus allowing identification of both the MF and AMF and distinguishing the latter from the nutritional orifices. This procedure should not be performed using panoramic radiography (OPG) since they give false negatives, because their image is not so sharp, making it almost imperceptible. These AMFs are imperceptible, having an efficiency of only 48%; in rare cases, these AMFs may be multiple [16].

The objective of this study was to recognize the presence of the double MF in the dry jaws of the Chilean population and to compile the greatest information on the clinical relevance reported in the literature.

## 2. Materials and Methods

In this descriptive observational study, we have collected fresh mandibles obtained from 3 private education institutions in Santiago, Chile, the morphology department at Andrés Bello University, the Normal Human Anatomy Unit of the University of Santiago, and the anatomy pavilion at the Faculty of Medicine of the Finis Terrae University. This study met the STROBE verification standards.

The samples for this research were obtained by convenience, and the observation of the jaws was carried out in the human anatomy laboratories of each institution by three evaluators independently (MO-JJV and PN) and a fourth evaluator who participated in the evaluation of the jaws to validate that each evaluation was carried out in the presence of an accessory MF. The jaws were all evaluated on the same day at each institution, since each evaluation did not present visual or cognitive alterations for each evaluator, as it is simple and quick. The data collected from the images were tabulated in an Excel software spreadsheet V. 2021 1.2.12 (Microsoft, 2020) by the same operator without the personal identification of each individual.

Data analysis was carried out through descriptive statistics using tables that described the number of jaws analyzed, the number of jaws with the presence of AMF, and the laterality of the variant. To establish if there was a relationship between the variable presence of the AMF in the sample studied in this research and with previous research, we have added the same data as in this research to make a comparison with previous data reported in other studies that were compared with the present study through a statistical analysis of Student’s *t*-test for means of two paired samples, with statistically significant values less than 0.05% (*p* = 0.05). For the analysis of the data reported by the previous studies, we have taken the mean and prevalence of each study, and we have made an average between all of them to then compare the mean difference with our study. Finally, for comparison with previous studies, 4 databases were reviewed, Medline, Scopus, WOS, and Google Scholar, and the following keywords were used: accessory MF, additional MF, double MF, anatomical variant, clinical considerations, and clinical anatomy.

## 3. Results

The study was designed using a descriptive approach to evaluate the presence of AMFs. This analysis was performed from an external view and by laterality in the mandible. Therefore, the prevalence of AMFs in the samples were reported, taking into account bilateral or unilateral presence, the proportion of occurrence on each side in unilateral cases, and the duplication of the foramen on each side. The samples for this research came from 260 dry jaws, which were obtained from three institutions in Santiago, Chile, showing the following findings of the total jaws studied, and in order to classify as an accessory MF, it was observed and measured so that it complied with what was declared in the literature as the presence of an AMF. In addition, we have added to each photograph an image that represents where the neurovascular bundle should go based on imaging studies that have reported this, which is between 0.74 mm and 0.89 mm, with which a total of 8 (3.07%) jaws were found that presented with an AMF, of which 5 were unilateral, 3 right (Figure 1), and 2 left (Figure 2), while 3 mandibles presented with an AMF bilaterally (Figure 3).

This information was compared with 19 similar studies using the same exclusion criteria presented in Table 1 [2,3,7,9,17,18,19,20,21,22,23,24,25,26,27,28,29,30,31] in order to compare the results and determine whether there was similarity with the data presented in Table 1, Chilean sample. Regarding the prevalence of AMFs, the studies were compared with the current study. First, the 17 included studies had samples that fluctuated between 1 and 4000, a cumulative total of 7946, and an average number of jaws analyzed in all of the studies of 467.4, which did not show statistically significant differences between the means with the sample analyzed in this study; *p* = 0.095. For the cumulative prevalence of the presence of AMFs, this was 3.07 in this study, and in the compared studies, the average of AMFs was 8.01%, which did not present a statistically significant difference; *p* = 0.158. Regarding the presence of variants of unilateral AMFs, this occurred in 5 jaws, which is equivalent to 1.84% in the samples of this study, while in previous studies, it was 7.5%, being higher on the left side than on the right side. In previous studies, it was greater on the left side. The presence of AMFs in this study occurred bilaterally in 1.24% compared to the average of previous studies, which was 0.51. In the present study, it was relatively higher but not statistically significant; *p* = 0.231. Finally, in the present study, the sex of the sample was not identified since the jaws were dry, and it would have been too complicated to identify the sex. On the other hand, previous studies that did report the sex of the sample showed that this particular variant occurred in 127 women, which is equivalent to 21.27% of the samples with the AMF variant, while 125 men presented the AMF variant, which is equivalent to 20.93% (Table 2).

## 4. Clinical Considerations

The recognition of AMFs contributes to the use of an adequate surgical technique and prevents possible damage to the nerves and vessels of the treated regions, a situation that is described by Torres, Zmyslowk, Paraskeva, Tiwari, Sun, Rayphema, and Munielo [2,15,20,21,29,31]. According to Savoldi [32,34], it is important to consider the use of three-dimensional CBCT images before surgical procedures in the area of the mandibular premolars and molars in order to evaluate the course of the AMF and the neurovascular structures in the area, preventing erroneous diagnoses and periosteal detachment during implant, periodontal, and periapical surgery, as described by Sun, Lam, Katamaki, Iwanaga, Gumusok, and Aljarbou [3,24,25,26,30]. Relating to the above, Naitoh [26] adds that more studies are needed on the interpretation of various fine neurovascular structures of the jaw, such as the mandibular bifid and mandibular lingual bone canals, using rotational panoramic radiographs where the information obtained from the CBCT is fed back. Savoldi and Guo [23,34] proposed that it could be useful to predict MFs in patients with missing teeth by looking at combined soft and hard tissue reference points. Furthermore, Iwanaga [24] suggests that the recognition of AMFs is useful for avoiding neurovascular complications during implant surgery, nerve blocks, and other oral surgery procedures. However, further general anatomical studies of AMNs and AMFs should be performed to clarify the courses of AMN function and allow for predictions about them. Iwanaga et al. [4] reported that, when the periosteum around the AMF is elevated, the small number of risk assessments should be taken into account to predict intraoperative and postoperative complications caused by damaging the foramen and accessory holes in the jaw that compromise the arteries and nerves in the area. During dental procedures, such as cleaning and modeling, doctors must respect the precise working length; otherwise, it could cause damage and/or side effects in the patient. This is mentioned in the study by Hester et al. [35], which states that excessive preparation of the canal and violation of the apical foramen, an opening where blood vessels and nerve endings pass that nourish the dental pulp (soft tissue found inside the tooth), can cause direct physical injury to the mental nerve and paresthesia therein. Consequently, if the doctor does not corroborate the presence of this anatomical variant, and if the patient presents with it, it could cause the damage mentioned above and also cause damage to an accessory branch of the nerve, overestimating the symptoms and complications of the treatment.

The situation mentioned above was evidenced in a study by Kqiku [36], which describes a patient who presents with paresthesia and anesthesia with swelling. Radiographically, the periapical lesion, which corresponds to a pathology at the level of the alveolar bone and represents an inflammatory response due to bacterial infection of the root canals, was very close to the mental nerve but without direct anatomical contact. They discussed various options as to what could have caused this; one possibility was local pressure on the mental nerve as a consequence of the accumulation of purulent exudate in the mandibular bone. It should be noted that, although the involvement is close to the dental alveolus joint, mostly the accessory MF is inferior. Studies have shown that, in the presence of an AMF, the normal MF could have a rise, which would cause this foramen and the neurovascular content to approach the treatment region, potentially causing alterations and vascular or nervous disorders associated with the procedure or treatment to be performed [36].

## 5. Discussion

In this observational study, we have verified the presence of MFs in dry jaws and shown that this structure in the Chilean population appears as a variant. We have also determined, through an analysis of the literature, that the MF presents important clinical considerations for surgeons.

Although we found no small number of studies that discuss AMFs, not all of them found their samples with the variant randomly; some studies evaluated only jaws with AMFs. The AMF is usually smaller than MF, ranging from 0.74 mm. to 0.89 mm. There is a distance between the AMF and MF of 0.67 mm to 6.3 mm; therefore, they can be located in different areas around the MF. Most commonly, the AMF is found inferior to the MF, either anterior or posterior. However, there have been cases where the AMF is located toward the superior between the first premolars and the second molar, but this location also depends on the study population, thus there are discrepancies among some studies. Naitoh [28] notes that AMFs occur mainly in the posteroinferior region of the MF. Balcioglu [5], on the other hand, presented a case of the AMF anterior to the MF in a 6-year- old child, and Han [37] reports that the AMF is located toward the anteroinferior of the MF. This could indicate that, depending on the ethnicity and population, different locations of the AMF could be associated, but there is no study with an adequate methodology that categorically confirms that the location is different according to ethnicity or race. Regarding anatomy and presence, this foramen tends to be confused with nutritional foramen, which can be multiple and are not connected to the mandibular canal, which is a substantial difference with the AMF, which has a close relationship with the mandibular canal and its neurovascular content. This confusion usually occurs when performing an orthopantomography, which can result in false negatives of the AMF due to the poor quality of the image, having more than 50% error when it is dictated as a nutritional foramen instead of the AMF. For this fact, a CBCT is the most appropriate way to search for this anatomical variation, because it provides a clear 3D image that will allow the doctor to see the differences that exist between the nutritional foramen concerning the AMF and MF.

On the other hand, the AMF has its origins in the branching of the mental nerve before it exits through the MF; embryologically, they complete their development in the 12th week of gestation after the branching of the inferior alveolar nerve. This means that both the AMF and MF become formed during similar time frames. Once the mandible matures, this foramen cannot form; therefore, an adult or pediatric patient who was born without one or more AMFs would not develop this anatomical variation in the future; that is, a single diagnosis would suffice. An example of this is described by Balcioglu [5], who presented the case of a 30-week fetus with an AMF on the right side. This could indicate the possibility of diagnosing this variation, either in a neonate or in an adult, thus decreasing possible risks in dental care across the entire age range.

Regarding the content, we have the inferior alveolar artery, inferior alveolar nerve, inferior alveolar vein, and other neurovascular structures. These structures innervate the gums, the teeth, the inner surface of the cheeks, the skin of the chin, the skin at the corners of the mouth, the mucous membrane, and the lower lip, in addition to irrigating the entire area surrounding these structures. Since they are all related to both the MF and AMF, it is important at a clinical level. An example of this is with anesthesia. To correctly carry out this process in a patient with an AMF, a double anesthesia must be performed, not only in the mental nerve of the MF but also in the AMF, through which this nerve will also pass. In turn, the accessory mental nerve (AMN) that passes through the AMF has branching patterns. Iwanaga [24] defines this distribution as being mainly directed toward the right angular region; however, Toh [33] defines three different patterns. The first being an AMF in the left zone to the MF; in this pattern, the AMN is a branch from the corner of the skin of the mouth that subsequently communicates with a branch of the facial nerve. The second pattern is a right AMF with a location superior to the MF; in this case, the AMN is a branch of the mucous membrane and skin belonging to the corner of the mouth that will subsequently generate communication with the oral nerve and innervate the gum of the molars. The third pattern is an AMF on the right side above the MF, whereby the AMN emerges from the AMF as a mucosal branch belonging to the medial area of the lower lip that will later communicate with branches of the inferior alveolar nerve. The branches vary by location—anterior, superior, posterior, and inferior—and by the distance between the AMF and MF, the latter being a determining factor. The second and third cases have the same location of the AMF but different distances between the MF and AMF; therefore, it is this distance added to the location that indicates how the AMN will be distributed since it complements and completes the innervation in areas where the MN (mental nerve) is not distributed.

In terms of its distribution, this varies according to ethnicity, population, and sex. There is a lower presence of these AMFs in the Caucasian population; for example, Russia has only a 1% prevalence, whereas there is a significant incidence in countries in Asia and the Middle East, including Japan, Sri Lanka, India, Iran, and Saudi Arabia, where the presence of AMFs is around 10% or more. Although authors such as Naitoh [22] indicate that there is no significant difference in prevalence according to gender, Toh [33] proposes that AMFs are more frequent in male patients of African or American ethnicity, and Gumusok [9] determined that the prevalence in Turkey was higher in men than in women. In that study, however, all multiple AMFs were in women, which indicates a possible tendency for women to have more than one AMF. These data show that this anatomical variant is common in many countries and populations, so it is important to consider it for the safety of patients and for a procedure that is both adequate and successful.

Regarding the symptoms in the studies presented, no sign has been described that causes any associated pathology; only the effects generated after a surgical procedure or through pathologies that affect surrounding structures are mentioned. Given the above, it could be deduced that patients could present this variant throughout their lives and be unaware of its presence since it does not produce symptoms by itself. In relation to clinical considerations, we have already mentioned that identifying this variant is of utmost importance, specifically to prevent damage to nerves and blood vessels associated with the MF and to avoid detachment of the periosteum during dental, periodontal, and periapical implant surgery, as it can present a different distribution from the established one. According to the above, the use of three-dimensional CBCT images before surgical procedures on the jaw is crucial to evaluate the course of nerves and blood vessels, preventing misdiagnosis and complications. Furthermore, neurovascular complications during dental procedures, such as implant surgery and nerve blocks, could be avoided by recognizing this anatomical variant. Also, in dental procedures such as cleaning and modeling, it is essential to respect the precise working length in cases where excessive preparation of the canal and violation of the apical foramen is performed on a person with this variant. Neglecting to check for the presence of this anatomical variant could cause direct physical injury to the mental nerve and paresthesia and could damage the accessory branch of the nerve, overestimating the symptoms and complications of the treatment. Paresthesia and anesthesia are some of the effects that are deduced according to the research that is generated by not carrying out an adequate review of the presence of this variant since these symptoms are exhibited by the presence of an infection or damage close to the mental nerve.

Finally, the data presented in this study show the similarities to previous studies in terms of the presence of AMFs in the laterality and the region where this variant occurs. We believe that identifying the anatomical variants is essential and useful in several fields, including medical practice. By identifying anatomical variants, doctors can recognize and diagnose diseases or medical conditions in a more specific and efficient way, avoiding errors and inadvertent injuries to important structures in surgical procedures.

## 6. Conclusions

The presence of AMFs is a variant with a high prevalence if we compare it with other variants of the jaw. Knowledge of the anatomy and position of the AMF is crucial to analyzing different scenarios in the face of surgical procedures or conservative treatments of the lower anterior dental region. When the presence of this variant is recognized, all precautions can be taken to avoid excessive bleeding or paresthesia or paresis of the mental nerve and its territory of innervation. Finally, we believe that the development of new clinical anatomical studies that associate the presence of the AMF will improve the management and treatment of these patients.

## Figures and Tables

**Figure 1 diagnostics-14-01277-f001:**
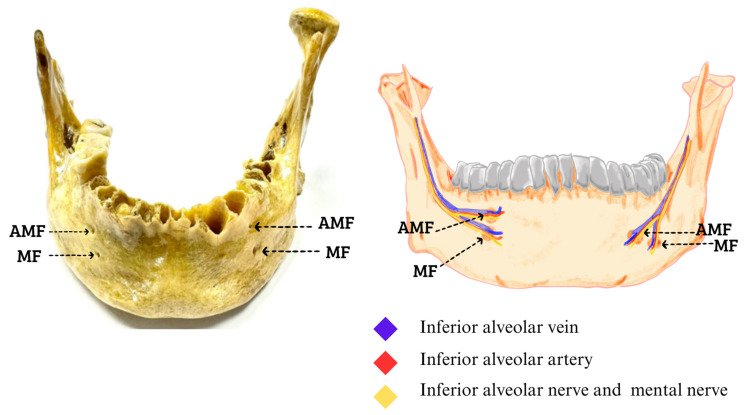
Bilateral accessory mental foramen. The superior AMF of both MFs; inside they pass the neurovascular bundle of the inferior alveolar vein, inferior alveolar artery, inferior alveolar nerve, and to the outside of the formants the mental nerve with its accessory branches that are directed towards the AMF (Figures made by the authors.).

**Figure 2 diagnostics-14-01277-f002:**
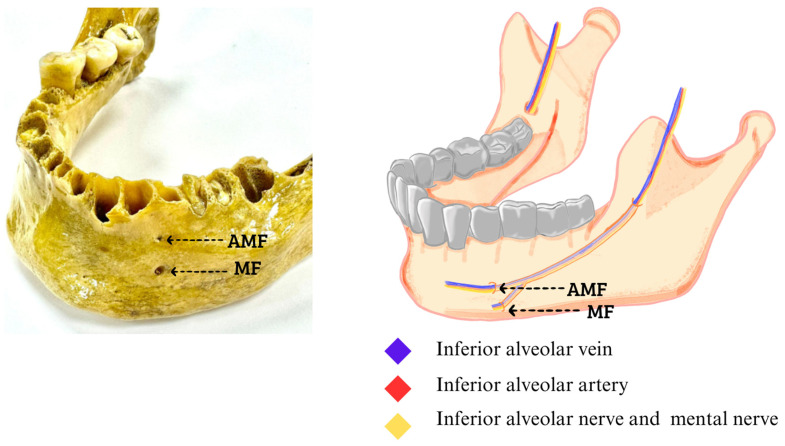
Left accessory mental foramen. The AMF at the top of the MF, inside the neurovascular bundle of the inferior alveolar vein, inferior alveolar artery, inferior alveolar nerve and to the outside of the formants the mental nerve with its accessory branches that are directed towards the AMF (Figures made by the authors.).

**Figure 3 diagnostics-14-01277-f003:**
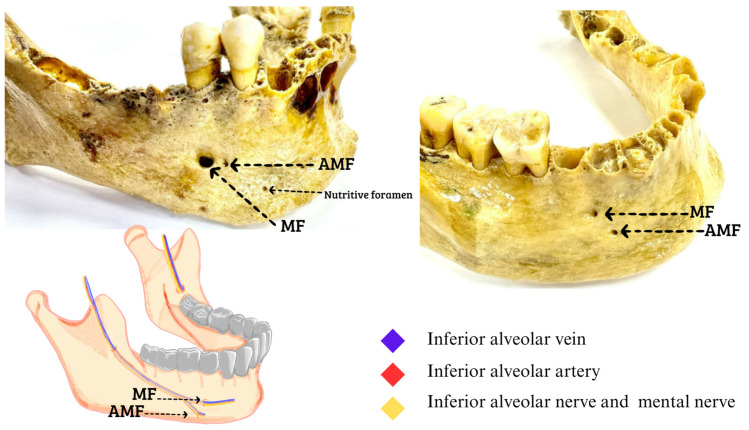
Right accessory mental foramen. The AMF at the bottom of the MF; inside the neurovascular bundle of the inferior alveolar vein, inferior alveolar artery, inferior alveolar nerve, and to the outside of the formants the mental nerve with its accessory branches that are directed towards the AMF. The rest of the foramina seen in the dry mandibles correspond to nutritional foramina (Figures made by the authors).

**Table 1 diagnostics-14-01277-t001:** Characteristics of the studies analyzed.

Author and Year	Country	Total N and Example	Presence Foramen Mentoniano Accesory	Unilateral (Left and Right)	Bilateral
Aljarbou, 2021 [3]	Saudi Arabia	603 via CBT	60/603 9.95%	27 left and 33 right	They did not report
Rahpeyma, 2018 [19]	Iran	5 via CBT and OPG	5/5 100%	1 left and 2 right	2
Zmysłowska-polakowska, 2017 [20]	Poland	200 via CBCT	28/200 10%	13 left and 15 right	They did not report
Torres, 2015 [21]	Brazil	1 via CBCT	1/1 100%	1 right	They did not report
Naitoh, 2011 [22]	Japan	365 via CBCT	28/365 7.67%	23 side is not specified	7
Gümüşok, 2017 [9]	Turkey	645 via CBCT	75/645 11.62%	69 side is not specified	6
Guo, 2009 [23]	China	21 through embalmed adult corpses	2/21 9.52%	2 side is not specified	They did not report
Iwanaga, 2016 [24]	Japan	63 via CBCT	9/63 14.28%	3 left and 1 right	5
Katakami, 2008 [32]	Japan	150 via CBCT	16/150 10.66%	15 side is not specified	1
Lam, 2019 [26]	Australia	4000 via CBCT	254/4000 6.35%	122 left and 110 right	12
Li, 2018 [27]	China	787 via CBCT	57/787 7.24%	16 left and 37 right	4
Muinelo Lorenzo, 2021 [31]	Spain	357 via CBCT and PAN	48/357 13.44%	20 left and 28 right	They did not report
Naitoh, 2009 [28]	Japan	157 via CBCT	15/157 9.55%	6 left and 9 right	They did not report
Paraskevás, 2015 [29]	Germany	96 through dry jaws	4/96 4.16%	4 side is not specified	They did not report
Sun, 2013 [30]	South Korea	446 via CBCT	39/446 8.74%	13 left and 20 right	6
Tiwari, 2022 [2]	Nepal	47 dry jaws	4/47 8.51%	4 left and 0 right	They did not report
Toh, 1992 [33]	Japan	3 corpses	3/3 100%	1 left and 2 right	They did not report
Current study, 2024	Chile	260 dry jaws	8/260 3.07%	3 left and 2 right	3

**Table 2 diagnostics-14-01277-t002:** Summary of articles included and compared with the current study.

Total Studies	Number of Jaws	Unilateral Prevalence	Bilateral Prevalence	Prevalence in Women	Prevalence in Men	Prevalence by Region
17	7946 Mean 467.4	597/7.5%	41/0.51%	127/21.27	125/20.93%	Asia: 12 Oceania: 1 Africa: 0 America: 1 Europe: 3
Current study, 2024	260	5/1.84%	3/1.23%	-	-	1 South America

## Data Availability

The data presented in this study are available on request from the corresponding author. The data are not publicly available due to privacy.

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
