# Peer review of "Morphological Characteristics of the Double Mental Foramen and Its Relevance in Clinical Practice: An Observational Study"

_diagnostics, 2024, doi:10.3390/diagnostics14121277_

Round 1
Reviewer 1 Report
Comments and Suggestions for Authors
Review to article_Morphological characteristics of the double mental foramen and its relevance in clinical practice. Observational study and literature review
1. The authors claimed that the study was realized on dried mandibles, so the trajectory of the neurovascular bundle of the inferior alveolar vein, inferior alveolar artery, inferior alveolar nerve presenting on the pictures is only a supposition. In conclusion, I believe that images showing the path of the vascular-nerve bundle should be removed.
2. The authors cannot compare the present study with studies in the literature in terms of gender prevalence, because as they themselves state, the study was performed on dry mandibles and it would have been too difficult to determine the gender. Table 2 shows the prevalence in women and man of the cases from the literature, but in discussions the authors support that ‘Regarding the sex of the patients, no studies have attributed the presence of AMF in a specific sex’. The authors should remove this sentence because on page 9 data from the literature on the prevalence of AMF according du gender are described.
3. I recommend the authors consider the study below:
- Yalcin TY, BektaÅŸ-Kayhan K, Yilmaz A, Ozcan I. An Alternative Classification Scheme for Accessory Mental Foramen. Curr Med Imaging. 2021;17(3):410-416. doi: 10.2174/1573405616666200928143014. PMID: 32988354.
4. the English can be improved:
- in sentence from rows 104-107 This procedure should not be done using a panoramic radiograph (OPG), as these give false negatives because their im- age is not as sharp, making it almost These AMFs are imperceptible, having an efficacy of only 48%; in rare cases, these AMFs can be multiple.[16]
- Row 139 MFA has to be replace with AMF
In conclusion I recommend minor review.
Author Response
Response reviewer 1
Dear, we appreciate your review and comments, since we are convinced that with the
suggested changes our study will improve, below I will detail the response to your
proposed comments:
Review to article “Morphological characteristics of the double mental foramen and its
relevance in clinical practice. Observational study and literature review”
1. The authors claimed that the study was realized on dried mandibles, so the trajectory of
the neurovascular bundle of the inferior alveolar vein, inferior alveolar artery, inferior
alveolar nerve presenting on the pictures is only a supposition. In conclusion, I believe that
images showing the path of the vascular-nerve bundle should be removed.
Response: It has been decided to look for studies that support the route of the
neurovascular bundle; these studies have been duly referenced.
2. The authors cannot compare the present study with studies in the literature in terms of
gender prevalence, because as they themselves state, the study was performed on dry
mandibles and it would have been too difficult to determine the gender. Table 2 shows
the prevalence in women and man of the cases from the literature, but in discussions the
authors support that ‘Regarding the sex of the patients, no studies have attributed the
presence of AMF in a specific sex’. The authors should remove this sentence because on
page 9 data from the literature on the prevalence of AMF according du gender are
described.
Response: We have taken your comment into consideration, therefore, it was decided to
delete the indicated text.
3. I recommend the authors consider the study below:
- Yalcin TY, BektaÅŸ-Kayhan K, Yilmaz A, Ozcan I. An Alternative Classification Scheme for
Accessory Mental Foramen. Curr Med Imaging. 2021;17(3):410-416. doi:
10.2174/1573405616666200928143014. PMID: 32988354.
Response: Thank you very much for the suggested study, we have considered it.
4. the English can be improved:
- in sentence from rows 104-107 This procedure should not be done using a panoramic
radiograph (OPG), as these give false negatives because their im- age is not as sharp,
making it almost These AMFs are imperceptible, having an efficacy of only 48%; in rare
cases, these AMFs can be multiple.[16]
- Row 139 MFA has to be replace with AMF
Response:
- It was considered to improve the suggested English wording, which was as follows: This
procedure should not be performed using panoramic radiography (OPG), since they give
false negatives because their image is not so sharp, making it almost imperceptible. These
AMFs are imperceptible, having an efficiency of only 48%; in rare cases, these AMFs may
be multiple.[16]
- MFA has been replaced by AMF.
Sincerely
Investigation group
Reviewer 2 Report
Comments and Suggestions for Authors
Dear Authors,
Thank You for a pleasure to read Your article.
I have several offers and comments to improve Your manuscript.
Title
The comparison of results of Your own research with other studies is obligate for most of scientific articles. There is no need to name the discussion with literature review.
Abstract
Please, add in this section the results of statistics analysis of data in Your study.
Key words: please, check them with MeSH and correct them.
Introduction
Lines 55-57: This 55 is relevant because the content of this AMF is the same as the main MF, that is, the inferior 56 alveolar artery, inferior alveolar nerve, mental nerve, and inferior alveolar vein.
Foramen could not contain any structures, but some structures could go through it. Please, correct this part.
Line 57: Among 57 other vessels and neurovascular structures.
What is the difference?
Materials and methods
Please, describe how exactly You performed all measures.
Describe statistics analysis.
Results
For figures, please, correct legends. They should contain the title, explanation of lines etc but not phrases as ‘You see’. Also, write, who is owner of these figures.
You could not compare results of anatomical study with x-ray methods, form the beginning they would have different results.
You have no results Your own research.
Please, Choose one of the article type and work with it (observational study or literature review).
Sincerely, Reviewer
Comments on the Quality of English LanguageThe article requires intensive English correction.
Author Response
Response reviewer 2
Dear, we appreciate your review and comments, since we are convinced that with the
suggested changes our study will improve, below I will detail the response to your
proposed comments:
Review to article “Morphological characteristics of the double mental foramen and its
relevance in clinical practice. Observational study and literature review”
Title
The comparison of results of Your own research with other studies is obligate for most of
scientific articles. There is no need to name the discussion with literature review.
Response: We have made the change you proposed.
Abstract
Please, add in this section the results of statistics analysis of data in Your study.
Response: We have added the suggested statistical data.
Key words: please, check them with MeSH and correct them.
Response: We have changed mental foramina to the term MeSh "Mental foramen".
Introduction
Lines 55-57: This 55 is relevant because the content of this AMF is the same as the main
MF, that is, the inferior 56 alveolar artery, inferior alveolar nerve, mental nerve, and
inferior alveolar vein.
Foramen could not contain any structures, but some structures could go through it.
Please, correct this part.
Line 57: Among 57 other vessels and neurovascular structures.
What is the difference?
Response: Response: We have added that the neurovascular bundle of the mental
foramen is divided in the distal portion of these structures, therefore, the content of the
mental foramen is identical to that of the accessory mental foramen.
Materials and methods
Please, describe how exactly You performed all measures.
Describe statistics analysis.
Response: We have considered your suggestions, which have already been changed in the
text.
Results
For figures, please, correct legends. They should contain the title, explanation of lines etc
but not phrases as ‘You see’. Also, write, who is owner of these figures.
You could not compare results of anatomical study with x-ray methods, form the
beginning they would have different results.
You have no results Your own research.
Please, Choose one of the article type and work with it (observational study or literature
review).
Response: Response: We have added the corresponding title to each figure and have
indicated their ownership. The results have been corrected based on the suggested
comment. We have added the results of our research at the end of Table 1. We chose
“observational study” in the title.
Sincerely
Investigation group